# Asymmetric Orthogonal Metasurfaces Governed by Toroidal Dipole Bound States in the Continuum

**Jun Ji, Xiaolong Lv, Chuanfei Li, Xiaoyuan Yang and Yunsheng Guo ***

School of Science, Inner Mongolia University of Science and Technology, Baotou 518060, China; lvxiaolongnm@163.com (X.L.); 2021023201@stu.imust.edu.cn (X.Y.)
* Correspondence: gys03018@imust.edu.cn

**Abstract:** An all-dielectric metasurface composed of orthogonal-slit silicon disks is proposed in this study. By modifying the unit structure of the metasurface with the bound states in the continuum (BICs), a sharp Fano resonance can be generated. The resonance properties of the metasurface are investigated by analyzing the effects of the structural parameters on the resonance using the eigenmode analysis method. The Q factor and the resonance wavelength can be adjusted by varying the slit width, the disk thickness, and the disk radius. The electromagnetic characteristics and mechanism of the toroidal dipole BICs (TD-BICs) are explored in depth through an analysis of the multipole expansion of the scattered power, along with the electromagnetic field and the current distribution at resonance. This research provides a novel approach for the excitation of strong TD-BIC resonance and proposes potential applications in optical switches, high-sensitivity optical sensors, and related areas.

**Keywords:** metasurface; all-dielectric metasurface; bound states in the continuum; TD-BICs; Fano resonance; Q factor

## 1. Introduction

In recent years, significant research attention has been focused on metasurfaces [1–7]. These surfaces are usually composed of artificial electromagnetic materials made from subwavelength-scale dielectric nanostructures. Metasurfaces have the ability to manipulate light efficiently and flexibly, thus bringing new design possibilities and methods to the optics field. All-dielectric metasurfaces in particular offer advantages that include low losses, high efficiency, and ease of integration, making these surfaces highly promising for a variety of applications, including optical imaging [8–11], optical information processing [12], and optoelectronic devices [13,14]. However, the limited nonlinear optical coefficients of traditional optical materials such as silicon have hindered the development progress of nonlinear optics in all-dielectric metasurfaces [15]. To enhance the nonlinear effects in these all-dielectric metasurfaces, one effective method is to use bound states in the continuum (BICs) to achieve high-Q resonances. Unlike conventional bound states, BICs are in a continuous spectrum, can coexist with extended waves, and remain completely bound without any radiation [16–20]. Based on BICs, a high level of localized electromagnetic field enhancement can be generated in all-dielectric metasurfaces, thus improving the efficiency of nonlinear processes significantly.

In addition, these metasurfaces can support Mie resonances [21,22], including electric dipole (ED), magnetic dipole (MD), and toroidal dipole (TD) resonances. A TD can be regarded as a circular arrangement of MDs or EDs that are compressed into a point and then connected head to tail, and an almost unknown third type of electromagnetic multipole, i.e., the toroidal multipole, is required, along with familiar magnetic multipoles and electric multipoles, to form a complete multipole representation of any radiating or nonradiating source [23,24]. It should be noted that the TD resonance in metal metasurfaces is relatively

weak, and it is often masked by stronger electric and magnetic multipoles [25,26]. However, the all-dielectric metasurface proposed in this paper enables the observation and detection of the TD response. By providing a reasonable design for the super-material unit structure, the generation of the TDs can be achieved. The TD resonance in metasurfaces provides new opportunities for the development of optical devices such as high-sensitivity sensors [27–29], and optical modulators and switches [30]. Furthermore, recent reports have revealed a strong association between the TD resonance and the BICs, allowing high Q and a strong electromagnetic field enhancement [31].

In this paper, we present the design of a novel silicon-based all-dielectric metasurface structure that breaks the in-plane symmetry of the metasurface structure by arranging high-resistivity silicon disks that contain a pair of orthogonal slits into clusters. Using this structure, the prominent feature of the BIC–Fano resonance can be observed in the transmission spectrum. Through the analysis of the multipole expansion of the microscopic scattered power and the electromagnetic field modulus distribution, we demonstrate the existence of a TD-BIC in this structure, and we also achieve the continuous tuning of the Fano resonance via the adjustment of the geometric parameters. This work provides a novel and effective scheme for designing nonlinear optics in all-dielectric metasurfaces.

## 2. Materials and Methods

Figure 1 shows the proposed all-dielectric metasurface, which is composed of resonant units consisting of slit silicon disks deposited on a quartz substrate (where the refractive index of silicon is n = 11.7). The silicon disks are distributed in a square lattice with a period of 100 μm (p). The disks have a radius of 30 μm (r), a thickness of 40 μm (t), and a gap width of 10 μm (d). The electromagnetic properties and the spectral response of the all-dielectric metasurface were numerically simulated using COMSOL Multiphysics 6.0 software, where periodic boundary conditions were applied on the four sides of the resonant unit, and perfectly matched layers were added at the top and bottom. The excitation field is an x-polarized plane wave propagating along the z-axis.

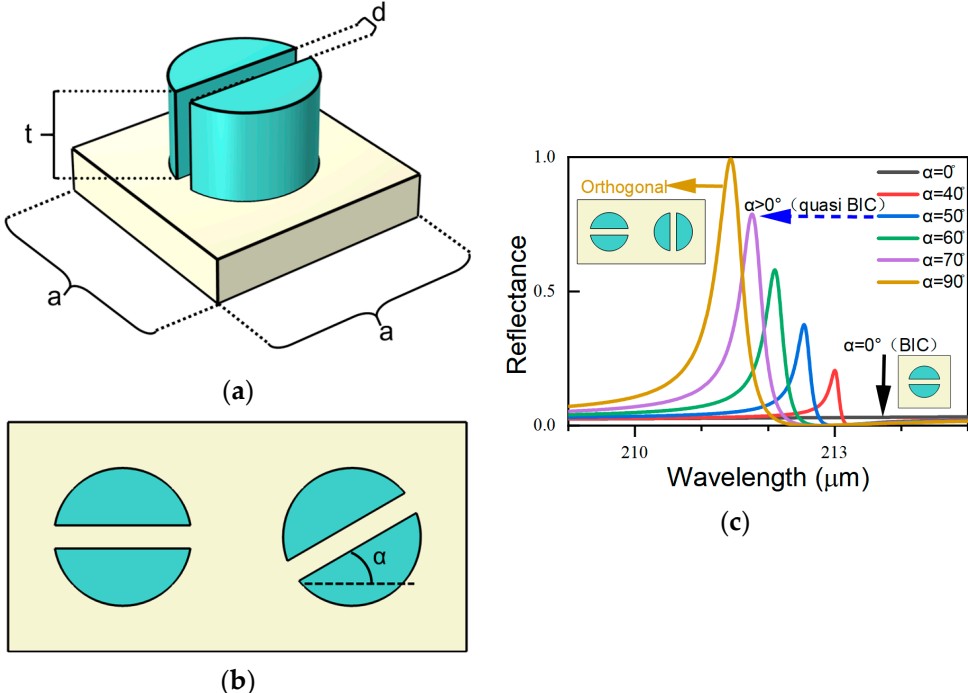

**Figure 1.** (**a**) The designed metasurface unit structure; (**b**) top view showing adjacent slits, where one is tilted at an angle $\alpha$; (**c**) reflection spectra at different values of the angle $\alpha$. When $\alpha = 0°$, the metasurface is in a bound state in the continuum (BIC). As $\alpha$ increases, the metasurface transforms from the BIC to quasi-BIC. When $\alpha = 90°$, the adjacent slits are orthogonally distributed.

### 3. Results

To initiate the BIC, we rotated adjacent silicon disk slits by an angle $\alpha$, thus breaking the in-plane symmetry of the metasurface. As $\alpha$ changes, the metasurface transforms from the BIC into quasi-BIC. When $\alpha = 0°$, the metasurface is in a BIC that cannot be excited by external fields and is unobservable. As $\alpha$ increases, a sharp asymmetric Fano line shape then appears and is accompanied by a blueshift in the resonance center and an increase in the modulation depth until $\alpha = 90°$, when the adjacent silicon disk slits are distributed orthogonally, and the modulation depth reaches a maximum. To ensure maximum reflectance, we studied the case of the orthogonal slit distribution. Figure 1c shows the reflection spectrum of the orthogonal distribution structure, as fitted using the Fano formula [32]:

$$T(\lambda) = T_0 + A_0 \frac{[q + 2(\lambda - \lambda_0)/\Gamma]^2}{1 + [2(\lambda - \lambda_0)/\Gamma]^2} \tag{1}$$

In the equation above, $q$ is the Fano fitting parameter that determines the asymmetry of the curve. $T_0$ and $A_0$ are coupling parameters. $\lambda_0$ represents the resonance peak wavelength and $\Gamma$ represents the resonance linewidth. Therefore, $Q = \frac{\lambda_0}{\Gamma}$. The resonance peak wavelength and resonance linewidth are 211.43 μm and 0.44 μm for this resonance, respectively, resulting in a Q factor of 480. To enable further quantitative analysis of the microscopic aspects of the multipole properties of the resonance, we also calculated the multipole expansion of the scattered power of this resonance in Cartesian coordinates [33,34].

Figure 2 illustrates the scattering powers of the ED, MD, TD, electric quadrupole (Qe), and magnetic quadrupole (Qm). From Figure 2, it is evident that the TD is dominant in resonance, followed by the Qm. The scattering power of the Qm is less than half that of the TD. Simultaneously, the ED, MD, and Qe are strongly suppressed. To explore the mechanism of this TD-BIC, we analyzed it from the perspective of its electromagnetic field and current profile. As shown in Figure 3, the displacement current forms two reversed loops in the x-z plane and the magnetic field forms a clockwise loop in the x-y plane, indicating that this is a TD resonance pattern along the negative z-direction. At the resonance wavelength, due to the presence of the dielectric layer, we can see that the electric and magnetic field enhancements can reach up to $2.74 \times 10^6$ and $3.1 \times 10^4$, far exceeding the case in free space.

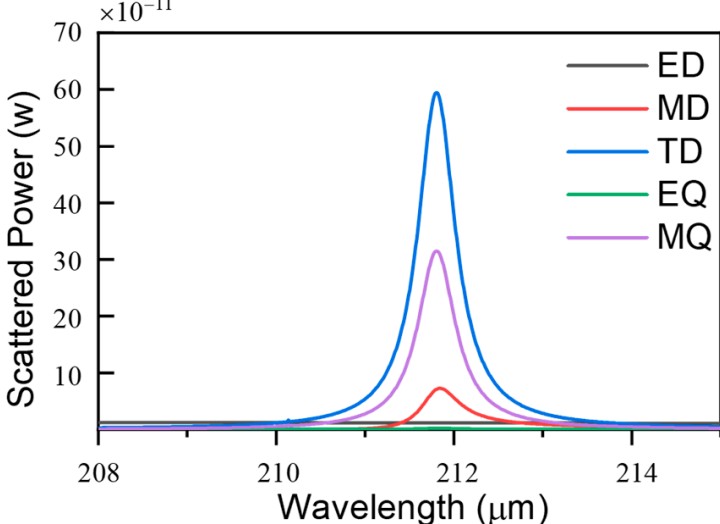

**Figure 2.** Multipole expansions of the scattered power.

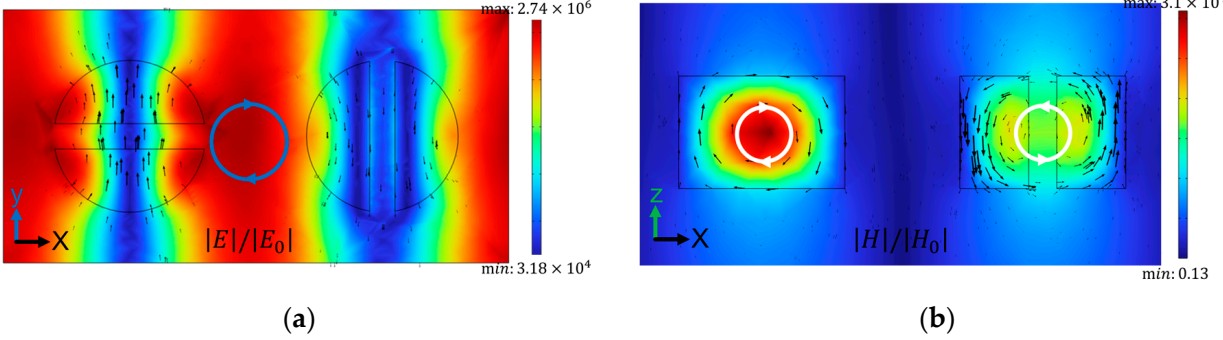

**Figure 3.** (**a**) Electric near-field enhancements in the x-y plane at the resonance wavelength, where the black arrows represent the magnetic field and form a counterclockwise magnetic field loop; (**b**) magnetic near-field enhancements and x-z plane at the resonance wavelength, where the black arrows represent the displacement current and form two reversed loops.

## 4. Discussion

We investigated the reflectance spectra of asymmetric metasurface structures with various geometric parameters. All geometric parameters were varied in steps of 1 µm, with only one parameter being varied in each figure, while the other parameters were maintained at their corresponding values, as shown in Figure 1. The results are shown in Figure 4. When the period, radius, and thickness of the unit structure were varied, a redshift in the resonance was observed. The linewidth slightly increased with increasing thickness. The redshift was more pronounced with increasing radius than it was with changing thickness, and increasing the radius also led to a larger linewidth and a reduced Q factor. The linewidth decreased, and the Q factor increased when the period increased. In contrast, increasing the gap width caused a blueshift in the resonance with almost no change observed in the linewidth. By comparing the effects of these different geometric parameters on the reflection spectra of the asymmetric metasurface structures, we found that the Fano resonance was more sensitive to radius than to the other parameters because the TD mode was a complex collective response that was more dependent on the radius than on the other geometric parameters. Based on these results, we can easily adjust the TD resonances of the asymmetric metasurface structures by changing the different geometric parameters. We set up a metasurface (p = 102 µm, r = 28 µm, t = 38 µm, and d = 12 µm) and achieved a Q factor of 1582. By further adjusting the geometric parameters, the Q factor can be increased.

We also studied the effect of the incident light's polarization angle on the resonance, and the results are shown in Figure 5. The results demonstrate that the reflection spectrum's modulation depth reaches a maximum when the incident light is polarized along the x direction, but the resonance disappears when the incident light is polarized along the y direction. This occurs because the designed metasurface structure only breaks the symmetry along the x direction, and periodic translational symmetry is maintained along the y direction. Optical switching may be achieved using this feature. When θ = 0°, electromagnetic waves at the resonance peak are reflected completely, and this state can be considered to be "off". When θ increases, the reflectivity then decreases, and this can be considered to be "on". The transmitted light intensity can then be modulated by simply changing θ.

In summary, we designed a TD-BIC metasurface structure in which the continuous tuning of the Fano resonance can be achieved by adjusting its geometric parameters. We revealed the mechanism of the TD-BIC by analyzing the multipole decomposition of the scattered power, along with the electromagnetic field and the polarization current. Additionally, our design and fabrication methods are universal and can easily be extended to different wavelengths. Furthermore, the designed metasurface structure can be applied to optical devices, including optical switches and high-sensitivity refractive index sensors.

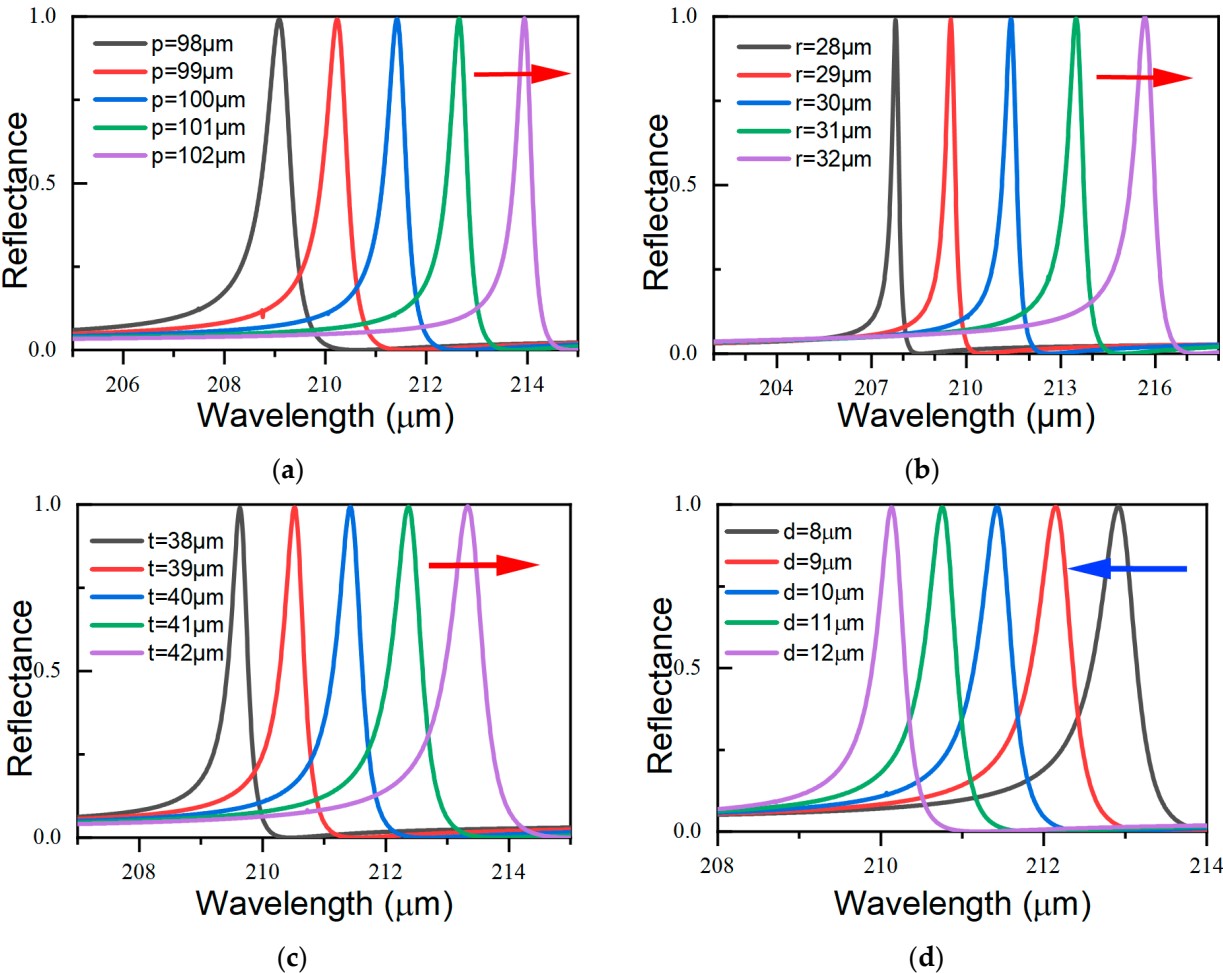

**Figure 4.** Reflection spectra of asymmetric metasurface structures with various values of (**a**) period (p), (**b**) radius (r), (**c**) thickness (t), and (**d**) slit width (**d**). The resonance caused a redshift with an increase in p, r, and t, and a blueshift with an increase in (**d**).

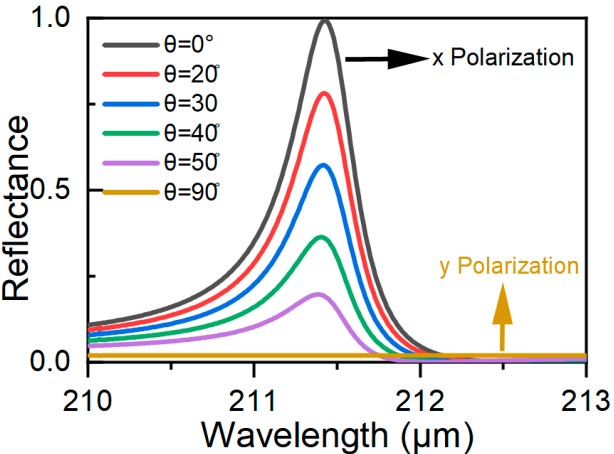

**Figure 5.** Reflection spectra of asymmetric metasurface structures with various values of the incident light polarization angle θ. The incident light is polarized along the x direction when θ = 0° and along the y direction when θ = 90°.

**Author Contributions:** Conceptualization, J.J.; methodology, J.J.; software, J.J.; validation, X.L. and X.Y.; formal analysis, J.J. and X.L.; investigation, J.J.; resources, X.L., C.L. and Y.G.; data curation, J.J.; writing—original draft preparation, J.J.; writing—review and editing, X.L. and Y.G.; visualization, J.J.; supervision, X.L., C.L. and Y.G.; project administration, X.L., C.L. and Y.G.; funding acquisition, X.L. and Y.G. All authors have read and agreed to the published version of the manuscript.

**Funding:** This research was funded by the National Natural Science Foundation of China (Grant Number 51862027) and the Natural Science Foundation of Inner Mongolia (Grant Number 2022QN01005).

**Institutional Review Board Statement:** Not applicable.

**Informed Consent Statement:** Not applicable.

**Data Availability Statement:** The data that support the findings of this study are available on reasonable request from the corresponding author. The data are not publicly available due to privacy.

**Conflicts of Interest:** The authors declare no conflict of interest.

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
