# Peer review of "Asymmetric Orthogonal Metasurfaces Governed by Toroidal Dipole Bound States in the Continuum"

_photonics, doi:10.3390/photonics10111194_

Round 1

Reviewer 1 Report

Comments and Suggestions for Authors

The authors proposed a dielectric metasurface and performed its superficial theoretical analysis. Unfortunately, just analyzing reflectance depending on certain parameters is definitely not enough to accept this article. In my opinion, this manuscript does not meet the criteria necessary to be published in Photonics.

Comments on the Quality of English Language

English is fine.

Author Response

Dear reviewer:

  While we appreciate the reviewer ' s feedback, we respectfully disagree. We think this study makes a valuable contribution to the field, and the reasons are as follows.

  This work introduces an innovative all-dielectric metasurface featuring orthogonal-slit silicon disks. Through modifications inspired by the bound states in the continuum (BICs) physics, it achieves a sharp Fano resonance. The study investigates the resonance characteristics by analyzing how structural parameters like slit width, disk thickness, and disk radius affect the Q factor and resonance wavelength. The research delves into the toroidal dipole-BIC (TD-BIC) and explores its electromagnetic properties and mechanisms. This metasurface design opens new possibilities for strong TD-BIC resonance, with potential applications in optical switches, high-sensitivity optical sensors, and related areas. It also highlights the significance of BICs in enhancing nonlinear effects in all-dielectric metasurfaces and introduces the TD resonance, which can be harnessed for optical devices.

  Overall, the manuscript is methodologically robust, with well-supported claims and conclusions. Consequently, this manuscript aligns with the scope of Photonics.

Reviewer 2 Report

Comments and Suggestions for Authors

Comments on the manuscript

This work introduces an innovative all-dielectric metasurface featuring orthogonal-slit silicon disks. Through modifications inspired by the bound states in the continuum (BICs) physics, it achieves a sharp Fano resonance. The study investigates the resonance characteristics by analyzing how structural parameters like slit width, disk thickness, and disk radius affect the Q factor and resonance wavelength. The research delves into the toroidal dipole-BIC (TD-BIC) and explores its electromagnetic properties and mechanisms. This metasurface design opens new possibilities for strong TD-BIC resonance, with potential applications in optical switches, high-sensitivity optical sensors, and related areas. It also highlights the significance of BICs in enhancing nonlinear effects in all-dielectric metasurfaces and introduces the TD resonance, which can be harnessed for optical devices.

In my assessment, the manuscript is interesting, and the theoretical/simulational findings contribute to BIC metasurfaces, a topic of high interest. Overall, the manuscript is methodologically robust, with well-supported claims and conclusions. Consequently, this manuscript aligns with the scope of Photonics, pending the resolution of a few minor concerns. Please find my suggestions and comments for the authors below.

To begin with, some typos. For example, in the abstract “An all-dielectric metasurface composed of orthogonal-slit silicon disks is proposed in this study. By modifying the unit structure of the metasurface with the bound states in the continuum (BIC) theory, a sharp Fano resonance can be generated.” Should be “bound states in the continuum (BICs)”, not “(BIC)”. Please revise or comment.

Also, this paper initially creates a negative impression as it lacks a thorough perspective in the first paragraph of the introduction section. For instance, it primarily focuses on dielectric BIC studies, while completely overlooking the recent advancements in plasmonic BICs. For example, quoting “Unlike conventional bound states, BICs are in a continuous spectrum, can coexist with extended waves, and remain completely bound without any radiation [13-16]”, it did not include any plasmonic examples. It is recommended to incorporate more comprehensive examples of recent progress in the realm of plasmonics, including: (1) anisotropic plasmonic BICs [Liang, Yao, et al. "Hybrid anisotropic plasmonic metasurfaces with multiple resonances of focused light beams." Nano Letters 21.20 (2021): 8917-8923; (2) the intriguing topic of chiral BICs in plasmonic metasurfaces.

Quoting “It should be noted that the TD resonance in metal metasurfaces is relatively weak and it is often masked by stronger electric and magnetic multipoles [21].” Recent advances in plasmonics TD/anapole (ED+TD) is missing. [Yezekyan, Torgom, et al. "Anapole states in gap-surface plasmon resonators." Nano Letters 22.15 (2022): 6098-6104].

Quoting, “Furthermore, recent reports have revealed a strong connection between the TD resonance and the BICs, allowing high Q and a strong electromagnetic field enhancement.” This claim lacks literature support.

I am continuing the last comment. This study is about lattice resonance where the resonance wavelength (~210μm) is close to the superlattice period (200μm). Usually, this kind of resonance is not known for its field enhancement but high Q-factor. How does the idea of “TD+BICs” contribute to strong field enhancement? This work does not show any evidence for strong field enhancement. Please comment. This question is related to the motivation of this work.

Comments on the Quality of English Language

readable

Author Response

请参阅附件。

Reviewer 3 Report

Comments and Suggestions for Authors

This paper is innovative and relevant, as it involves the advanced research fields of metasurfaces, Fano resonance, and TD-BIC. The paper structure is clear and logical, and the abstract presents the research purpose, method, and results, following the scientific writing standards. The method is reasonable and effective, as the multipole expansion method can quantify the contribution of the scattered power, and the electromagnetic field and current distribution diagrams can visually show the characteristics of TD-BIC. The results are meaningful and convincing, and also provide some potential application scenarios, demonstrating the functionality of the metasurface.

However, I have the following two questions that need further clarification from you:

1. What material did you use for the substrate of your metasurface structure? How does this affect your results?
    2. In your paper, you mentioned that the quality factor of a metasurface structure is 480. You also mentioned that by changing the structural parameters of the metasurface, the quality factor can be improved. However, you did not give an estimate of how much it could be improved. I suggest that you provide an estimated value or a possible range, which will give readers a clearer understanding.

Round 2

Reviewer 1 Report

Comments and Suggestions for Authors

After carefully analyzing the authors' responses, I maintain my opinion on the rejection of the manuscript. The proposed topic is well known, there are dozens of similar articles. I have listed examples below:

>Wang, Wudeng, et al. "High Q-factor multiple Fano resonances for high-sensitivity sensing in all-dielectric metamaterials." OSA Continuum 2.10 (2019): 2818-2825.

>Li, Meiqi, et al. "Multiple toroidal dipole symmetry-protected bound states in the continuum in all-dielectric metasurfaces." Optics & Laser Technology 154 (2022): 108252.

>Long, Xiyu, et al. "Sharp Fano resonance induced by all-dielectric asymmetric metasurface." Optics Communications 459 (2020): 124942.   The proposal of a metasurface does not bring anything new to the physics of the phenomenon, and the work is purely theoretical.